

# Parity and the modular bootstrap

**Tarek Anous[1] [⋆], Raghu Mahajan[2,3] [†] and Edgar Shaghoulian[4] [‡]**

**1** Department of Physics and Astronomy,
University of British Columbia, Vancouver, BC V6T 1Z1, Canada
**2** Department of Physics, Princeton University, Princeton, NJ 08540, USA
**3** School of Natural Sciences, Institute for Advanced Study, Princeton, NJ 08540, USA
**4** Department of Physics, Cornell University, Ithaca, NY 14853, USA

⋆ tarek@phas.ubc.ca † raghu_m@princeton.edu ‡ eshaghoulian@cornell.edu

## Abstract

We consider unitary, modular invariant, two-dimensional CFTs which are invariant under the parity transformation $P$. Combining $P$ with modular inversion $S$ leads to a continuous family of fixed points of the $SP$ transformation. A particular subset of this locus of fixed points exists along the line of positive left- and right-moving temperatures satisfying $\beta_L \beta_R = 4\pi^2$. We use this fixed locus to prove a conjecture of Hartman, Keller, and Stoica that the free energy of a large-$c$ CFT$_2$ with a suitably sparse low-lying spectrum matches that of AdS$_3$ gravity at all temperatures and all angular potentials. We also use the fixed locus to generalize the modular bootstrap equations, obtaining novel constraints on the operator spectrum and providing a new proof of the statement that the twist gap is smaller than $(c-1)/12$ when $c > 1$. At large $c$ we show that the operator dimension of the first excited primary lies in a region in the $(h, \overline{h})$-plane that is significantly smaller than $h + \overline{h} < c/6$. Our results for the free energy and constraints on the operator spectrum extend to theories without parity symmetry through the construction of an auxiliary parity-invariant partition function.



# 1 Introduction

Discrete symmetries play an important role in many modern developments in quantum field theory, appearing in mixed 't Hooft anomalies [1–22], conformal field theories on nontrivial backgrounds [23–29], and quantum gravity [30–49]. In this note we illustrate the conceptual and technical utility of parity symmetry in two-dimensional conformal field theory through two examples. First, we prove a conjecture by Hartman, Keller, and Stoica (HKS) [32] fixing the free energy at finite temperature *and* angular potential in large-$c$ CFT$_2$'s with a sparse light spectrum. Second, we generalize the modular bootstrap to theories with parity symmetry. When $c > 1$, we obtain new bounds on the first excited primary and give a new proof of the twist gap being at most $(c-1)/12$. Both results apply to theories without parity symmetry, since we only utilize the symmetry through the construction of an auxiliary parity-invariant partition function $\widetilde{Z}(\tau, \overline{\tau}) = Z(\tau, \overline{\tau}) + Z(-\overline{\tau}, -\tau)$. Results derived for $\widetilde{Z}(\tau, \overline{\tau})$ can be shown to apply to $Z(\tau, \overline{\tau})$ as well.

**Background**

The modular bootstrap, first considered by Cardy [50], is a "medium-temperature" expansion (about $\beta = 2\pi$) of CFT$_2$ observables. The basic idea is to impose invariance of the partition function under the $S$ transformation

$$S: \quad Z(\tau, \overline{\tau}) \mapsto Z\left(-\frac{1}{\tau}, -\frac{1}{\overline{\tau}}\right). \tag{1.1}$$

Invariance under $S$ implies the vanishing of certain derivatives of $Z$ at the fixed point of this transformation $(\tau, \overline{\tau}) = (i, -i)$.

In [51], Hellerman rediscovered the modular bootstrap and used it to prove an upper bound on the scaling dimension of the first excited Virasoro primary $\Delta^{(1)}$. Hellerman's bound states that $\Delta^{(1)} < c/6$ at large $c$. In the language of AdS$_3$ gravity, $c/6$ is the dimension of the lightest BTZ black hole which dominates the canonical ensemble, at $\beta = 2\pi - 0^+$. However, BTZ black holes are known to exist all the way down to $\Delta = c/12$, suggesting it should be possible to improve the upper bound on $\Delta^{(1)}$ to $c/12$. So far, any such attempt has involved going to higher order in the expansion around the fixed point, e.g. [52, 53]. While this strategy shows improvement at finite $c$, none have successfully improved the bound at asymptotically large $c$. New tools may be necessary.

A possible generalization of the modular bootstrap is to use fixed points of other transformations of the full modular group $SL(2, \mathbb{Z})$. This means supplementing $S$ invariance with $T$ invariance $Z(\tau, \overline{\tau}) = Z(\tau + 1, \overline{\tau} + 1)$. For example, reference [54] considered the transforma-

tion $ST$.[1] However, individual terms in the partition function evaluated at such fixed points are not positive. This precludes applying ordinary bootstrap techniques around these $SL(2,\mathbb{Z})$ fixed points.

In this paper, we make some progress by constructing an auxiliary, parity-invariant partition function that is amenable to standard bootstrap techniques.

**Structure of the paper**

In section 2, we show that combining parity symmetry $P$ with modular inversion $S$ leads to a continuous family of fixed points of the $SP$ transformation. This continuous family of fixed points contains points with arbitrary $\beta \in \mathbb{R}^+$. This generalizes the $S$ fixed point, which sits at $\beta = 2\pi$. The partition function along this fixed locus is manifestly non-negative, allowing us to employ standard bootstrap techniques.

In section 3, we provide a proof of the HKS conjecture [32]. Our proof relies on $SP$ invariance of the auxiliary quantity $\widetilde{Z}(\tau, \overline{\tau}) = Z(\tau, \overline{\tau}) + Z(-\overline{\tau}, -\tau)$.

In section 4, we generalize the modular bootstrap equations of [51] to the case of $SP$-invariant partition functions. These equations can be evaluated at any generic fixed point along the line $\beta_L \beta_R = 4\pi^2$, i.e. $\beta = (\beta_L + \beta_R)/2 \geq 2\pi$, where the positivity of the partition function is manifest. With these new bootstrap equations, we derive novel constraints on the partition function and the spectrum of primary operators. For example, we show that it is possible to shrink Hellerman's triangular region $h + \overline{h} < c/6 + O(1)$ quite significantly, i.e. by $O(c)$ amounts. However, we cannot improve the bound on $h + \overline{h}$ at large $c$; see figure 2. We also give a new proof of the statement that the twist gap is at most $(c-1)/12$ when $c > 1$.

Section 5 contains brief remarks on related topics. Kinematic definitions are collected in appendix A.

# 2 Modular invariance plus parity

The parity transformation acts on the partition function as follows (see appendix A for definitions)

$$P: \quad Z(\tau, \overline{\tau}) \mapsto Z(-\overline{\tau}, -\tau) \,. \tag{2.1}$$

We remind the reader that the partition function is a function of independent complex variables $\tau$ and $\overline{\tau}$. Let us justify the action of parity (2.1). Physically, parity maps a state with energy $E$ and spin $J$ to a state with energy $E$ and spin $-J$. Thus, on the section $\overline{\tau} = \tau^*$, parity changes the sign of the angular potential. In particular, the partition function is invariant under $P$ if and only if the density of states $\rho(E, J)$ satisfies $\rho(E, J) = \rho(E, -J)$. Equation (2.1) is the natural holomorphic upliftment of this to $\mathbb{C}^2$.

From (1.1) and (2.1), we see that the combination $SP$ transforms the partition function as

$$SP: \quad Z(\tau, \overline{\tau}) \mapsto Z\left(\frac{1}{\overline{\tau}}, \frac{1}{\tau}\right). \tag{2.2}$$

The fixed locus of this transformation is $\tau \overline{\tau} = 1$, which is a complex curve (and hence a real 2-surface) inside $\mathbb{C}^2$.

---

[1]We know that $T$ invariance is satisfied if and only if spins are integer quantized. Thus, imposing $SL(2,\mathbb{Z})$ invariance around a more general $SL(2,\mathbb{Z})$ fixed point for a spectrum with only integer spins is equivalent to imposing $S$ invariance. However, different fixed points will package the same constraints in different ways; a given constraint may be easier or harder to access depending on the fixed point chosen.

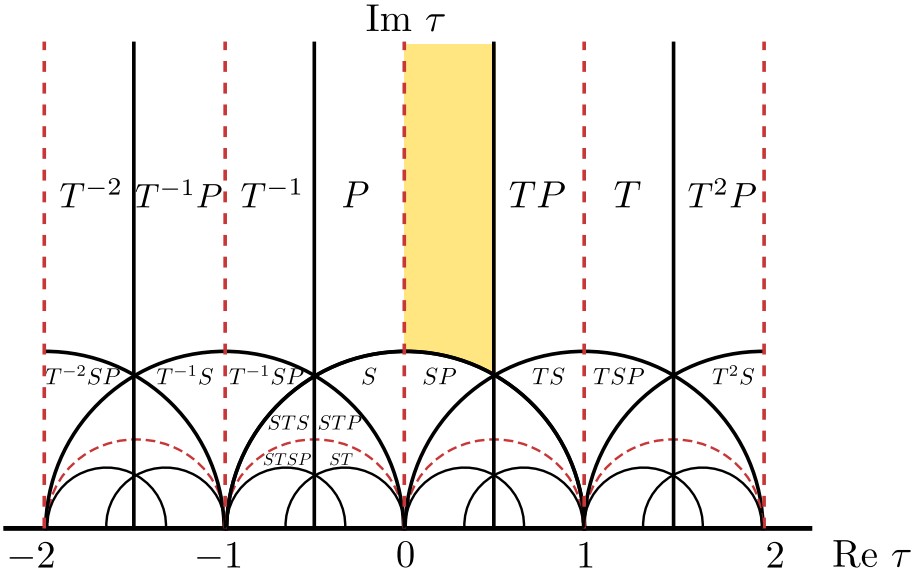

Figure 1: Fundamental domain and its images for $SL(2, \mathbb{Z})$ extended by parity $P$. New boundaries arising from the parity transformation are delineated by the dashed red curves.

Since $P$ and $S$ both square to the identity and commute with one another, $SP$ is the only nontrivial transformation incorporating both. Altogether, $S$, $P$, and $T$ can be shown to generate the following general transformations:

$$Z(\tau, \overline{\tau}) \mapsto Z\left(\frac{a\tau + b}{c\tau + d}, \frac{a\overline{\tau} + b}{c\overline{\tau} + d}\right) \quad \text{or} \quad Z(\tau, \overline{\tau}) \mapsto Z\left(\frac{-a\overline{\tau} + b}{-c\overline{\tau} + d}, \frac{-a\tau + b}{-c\tau + d}\right), \tag{2.3}$$

where, as usual $a, b, c$ and $d$ are integers such that $ad - bc = 1$. The refinement of the fundamental domains of the modular parameter $\tau$ for theories with parity symmetry is illustrated in figure 1. Notice from the figure that the union of any fundamental domain with its image under $P$ fills out an $SL(2, \mathbb{Z})$ fundamental domain. This means that to get the most general transformation which includes parity, we need to act with parity just once. This justifies the fact that (2.3) exhausts all possible transformations.

**Slices of the $SP$ fixed locus**

We now consider various slices of the $SP$ fixed locus, i.e. the curve $\tau\overline{\tau} = 1$. First, let us parametrize the partition function in terms of $\beta$ and $\theta$. Using the definitions in (A.1) and (2.2), $SP$ invariance implies that

$$Z(\beta, \theta) = Z\left(\frac{4\pi^2\beta}{\beta^2 + \theta^2}, \frac{4\pi^2\theta}{\beta^2 + \theta^2}\right), \tag{2.4}$$

which has a fixed locus along the curve $\beta^2 + \theta^2 = 4\pi^2$. Restricting to real $\beta$ and $\theta$ along this locus, we have access to temperatures $\beta \leq 2\pi$, which includes the infinite-temperature limit $\beta \to 0$. We remark that the partition function for real $(\beta, \theta)$ is generically complex, but is real if and only if the theory is parity invariant. However, the individual terms in the partition function are not positive definite.

If we instead take $\beta_L, \beta_R$ as independent variables, the statement of $SP$ invariance becomes (again using (A.1) and (2.2))

$$Z(\beta_L, \beta_R) = Z\left(\frac{4\pi^2}{\beta_R}, \frac{4\pi^2}{\beta_L}\right), \tag{2.5}$$

with fixed locus $\beta_L \beta_R = 4\pi^2$. From (A.2), we see that restricting to real $\beta_L$ and $\beta_R$ along this slice allows us to access temperatures with $\beta \geq 2\pi$, including the zero-temperature limit $\beta \to \infty$. We can reach this limit either by taking $\beta_L \to \infty$ or $\beta_R \to \infty$. The partition function is manifestly real and positive along the real $(\beta_L, \beta_R)$ slice: $Z(\beta_L, \beta_R) = \sum_{E_L, E_R} \rho(E_L, E_R) \exp[-\beta_L E_L - \beta_R E_R]$. This will be of crucial importance later.

The only point along the curve $\beta_L \beta_R = 4\pi^2$ that is also an $SL(2, \mathbb{Z})$ fixed point is $(\beta_L, \beta_R) = (2\pi, 2\pi)$. This is because $T$ acts as an imaginary shift of the left and right moving temperatures

$(\beta_L, \beta_R) \to (\beta_L - 2\pi i, \beta_R + 2\pi i)$. Thus we see that partition functions invariant under $SP$ exhibit new fixed points where the partition function is manifestly real and positive.

Altogether, by considering either the real $(\beta_L, \beta_R)$ slice or the real $(\beta, \theta)$ slice, we have access to fixed points at any temperature $\beta \in \mathbb{R}^+$. Henceforth, we restrict our attention to the slice with $\beta_L$ and $\beta_R$ both real.

# 3 CFT derivation of AdS$_3$ free energy

HKS [32] proved two important results for the free energy of CFT$_2$ at finite temperature and zero angular potential. First, they showed that if the central charge $c$ is large, the free energy at *all* temperatures is completely fixed by the spectrum of light states ($\Delta < c/12$) only. Second, assuming that the light spectrum is sparse ($\rho(\Delta) \lesssim e^{2\pi\Delta}$) in addition to large $c$, they were able to reproduce the free energy of Einstein gravity in AdS$_3$.

HKS also provided evidence that a slightly stronger sparseness condition on the low-lying spectrum fixes the free energy at nonzero angular potential. Ultimately however, they were unable to prove this statement, leaving it as an open conjecture. In this section we will give a proof of their conjecture.

## 3.1 Proof of the HKS conjecture

Let $\epsilon$ be a small positive number. Let us define *heavy* states to be the states which have $h > c/24 + \epsilon$ and $\overline{h} > c/24 + \epsilon$,

$$H = \{(h, \overline{h}) | h > c/24 + \epsilon \text{ and } \overline{h} > c/24 + \epsilon\}. \tag{3.1}$$

The remaining set of states is denoted by $H^c$, which contains states with low twist. Here the superscript $c$ is a mnemonic for set complement. Let $Z_H$ denote the partition function restricted to the states in $H$, and $Z_{H^c}$ the partition function restricted to the states in $H^c$.

Consider temperatures $\beta_L$, $\beta_R$ such that $\beta_L \beta_R > 4\pi^2$. Let $\beta_L'$ and $\beta_R'$ denote the $SP$ transformed temperatures. Explicitly, $\beta_L' := 4\pi^2/\beta_R < \beta_L$ and $\beta_R' := 4\pi^2/\beta_L < \beta_R$. For brevity we will denote by $Z$ the partition function evaluated at $(\beta_L, \beta_R)$, and by $Z'$ the partition function evaluated at $(\beta_L', \beta_R')$. The $SP$ invariance of the partition function implies $Z = Z_H + Z_{H^c} = Z_H' + Z_{H^c}' = Z'$, which is the same as

$$Z_H' - Z_H = Z_{H^c} - Z_{H^c}'. \tag{3.2}$$

Using the fact that the $SP$ transformed temperatures are smaller, we can immediately get an upper bound on $Z_H$:

$$Z_H = \sum_{(E_L, E_R) \in H} e^{-\beta_L E_L - \beta_R E_R} \leq \exp[\epsilon(\beta_L' - \beta_L + \beta_R' - \beta_R)] Z_H'. \tag{3.3}$$

Let us define $\varepsilon := \exp[\epsilon(\beta'_L - \beta_L + \beta'_R - \beta_R)]$. Note that $0 < \varepsilon < 1$. Now we massage equation (3.3) into

$$Z'_H \leq \frac{Z'_H - Z_H}{1 - \varepsilon} = \frac{Z_{H^c} - Z'_{H^c}}{1 - \varepsilon} \leq \frac{Z_{H^c}}{1 - \varepsilon} . \tag{3.4}$$

Here the first inequality is a direct consequence of (3.3), the second equality uses $SP$ invariance (3.2), and the third simply uses the positivity of $Z'_{H^c}$. Combining (3.3) and (3.4) we find:

$$Z_H \leq \frac{\varepsilon}{1 - \varepsilon} Z_{H^c} . \tag{3.5}$$

What the last inequality achieves is an upper bound on $Z_H$, the contribution of the heavy states to the partition function, in terms of the contribution of the non-heavy states. This immediately implies that we can also place an upper bound on the full partition function:

$$Z_{H^c} \leq Z \leq \frac{Z_{H^c}}{1 - \varepsilon} . \tag{3.6}$$

The first inequality comes from positivity of $Z_H$ and the second one follows from writing $Z = Z_H + Z_{H^c}$ and using (3.5). An identical analysis for the case $\beta_L \beta_R < 4\pi^2$ bounds $Z$ by $Z'_{H^c}$. Notice that the upper bound goes to infinity as $\varepsilon \to 1$, which occurs as $\beta_L \beta_R \to 4\pi^2$. Thus, as long as we keep both $\beta_L$ and $\beta_R$ fixed (with $\beta_L \beta_R \neq 4\pi^2$) as we scale $c \to \infty$, we obtain

$$\log Z = \begin{cases} \log Z_{H^c} + \mathcal{O}(1) & \text{if } \beta_L \beta_R > 4\pi^2 \\ \log Z'_{H^c} + \mathcal{O}(1) & \text{if } \beta_L \beta_R < 4\pi^2 \end{cases} . \tag{3.7}$$

The $\log Z_{H^c}$ and $\log Z'_{H^c}$ pieces are $\mathcal{O}(c)$ due to the contribution of the vacuum. This shows that the free energy is dominated by the non-heavy states at large $c$. To match the bulk AdS$_3$ gravity answer,

$$\log Z(\beta_L, \beta_R) = \begin{cases} \frac{c}{24}(\beta_L + \beta_R) & \beta_L \beta_R > 4\pi^2 \\ \frac{c}{24}\left(\frac{4\pi^2}{\beta_R} + \frac{4\pi^2}{\beta_L}\right) & \beta_L \beta_R < 4\pi^2 \end{cases} , \tag{3.8}$$

we want $\log Z_{H^c}$ to be given solely by the vacuum contribution up to $\mathcal{O}(1)$ corrections for $\beta_L \beta_R > 4\pi^2$ (and similarly for $\log Z'_{H^c}$). This will be assured if we assume that $\rho(h, \bar{h}) \lesssim \exp[\beta_L h + \beta_R \bar{h}]$ for all $(h, \bar{h}) \in H^c$ and for all $(\beta_L, \beta_R)$ such that $\beta_L \beta_R \geq 4\pi^2$. Here, $\lesssim$ means that one is allowed to have polynomial prefactors. In turn this condition is guaranteed if

$$\rho(h, \bar{h}) \lesssim \exp\left[4\pi \sqrt{h\bar{h}}\right] \quad \text{for all } (h, \bar{h}) \in H^c . \tag{3.9}$$

Thus, we have shown that (3.9) implies the AdS$_3$ free energy (3.8) in a large-$c$ theory. The reverse implication can also be proven as long as we modify the bound for states that do not have $h = \mathcal{O}(c)$ and $\bar{h} = \mathcal{O}(c)$. We will omit the details here, but the essence of the argument is the same: bound the density of states around any such state so that it does not contribute at the same order as the vacuum answer.

Remarkably, we can now show the results in this section hold if we relax the requirement that the CFT be parity invariant. Starting with a possibly non parity symmetric $Z(\beta_L, \beta_R)$, construct the parity invariant (and still modular invariant) object

$$\widetilde{Z}(\beta_L, \beta_R) := Z(\beta_L, \beta_R) + Z(\beta_R, \beta_L) = \sum \widetilde{\rho}(h, \bar{h}) e^{-\beta_L(h - c/24) - \beta_R(\bar{h} - c/24)} . \tag{3.10}$$

If $\rho$ satisfies (3.9), $\widetilde{\rho}$ also satisfies (3.9) since

$$\widetilde{\rho}(h, \bar{h}) = \rho(h, \bar{h}) + \rho(\bar{h}, h) \leq 2 \max\left[\rho(h, \bar{h}), \rho(\bar{h}, h)\right] . \tag{3.11}$$

Thus $\widetilde{Z}(\beta_L, \beta_R)$ is given by (3.8). Next note that $\log Z_{\text{vac}} \leq \log Z \leq \log \widetilde{Z}$. Since (3.8) says that $\log \widetilde{Z} \approx \log Z_{\text{vac}}$, we see that $\log Z$ is both upper and lower bounded by the same quantity, and hence must be equal to it.

## 3.2 Cardy formula for $\rho(E_L, E_R)$

Like in [32], the above universal form of the free energy (3.8) allows us to conclude that the Cardy formula for the density of states extends beyond the regime of validity of Cardy's saddle point derivation. We can get the microcanonical density of states by inverse-Laplace transforming (3.8):

$$\rho(E_L, E_R) = \int_{\gamma-i\infty}^{\gamma+i\infty} d\beta_L d\beta_R \, Z(\beta_L, \beta_R) \, e^{\beta_L E_L + \beta_R E_R} \,, \tag{3.12}$$

where $\gamma > 0$. We can do the integral by saddle point. If $\sqrt{E_L E_R} > c/24$, this saddle point is at a value of $\beta_L$ and $\beta_R$ such that $\beta_L \beta_R < 4\pi^2$. We thus get the Cardy formula

$$\rho(E_L, E_R) = \exp\left(2\pi\sqrt{\frac{c}{6}E_L} + 2\pi\sqrt{\frac{c}{6}E_R}\right) \,, \qquad \text{if } \sqrt{E_L E_R} > c/24 \,. \tag{3.13}$$

Note that this formula is valid for all $\sqrt{E_L E_R} > c/24$, and not just for values of $E_L, E_R$ that are much bigger than $c$. This gives a CFT derivation of the Bekenstein-Hawking entropy for all rotating BTZ black holes which dominate the canonical ensemble.

Orthogonally to the methods spelled out in this section, reproducing the AdS$_3$ free energy at finite temperature and angular potential can be achieved by assuming a particular center symmetry breaking pattern [39].

## 4 The parity-modular bootstrap

Now we turn to deriving constraints on the operator spectrum.

### 4.1 New bootstrap constraints

Let us work in the real $(\beta_L, \beta_R)$ plane. Recall that invariance under $S$ implies that $Z(\beta_L, \beta_R) = Z\left(4\pi^2/\beta_L, 4\pi^2/\beta_R\right)$. The fixed point of this transformation is at $\beta_L = \beta_R = 2\pi$. This implies

$$(\beta_L \partial_{\beta_L})^{N_L} (\beta_R \partial_{\beta_R})^{N_R} Z(\beta_L, \beta_R)\big|_{\beta_L = \beta_R = 2\pi} = 0 \,, \quad \text{for } N_L + N_R \text{ odd.} \tag{4.1}$$

This was used in [51] to derive an upper bound on the dimension $\Delta^{(1)}$ of the lightest primary in a modular invariant 2d CFT.

Generalizing this to $SP$ invariant theories, we get that along the fixed locus $\beta_L \beta_R = 4\pi^2$ the following derivatives vanish:

$$\partial_{(N_L, N_R)} := (\beta_L \partial_{\beta_L})^{N_L} (\beta_R \partial_{\beta_R})^{N_R} - (-1)^{N_L + N_R} (\beta_L \partial_{\beta_L})^{N_R} (\beta_R \partial_{\beta_R})^{N_L}$$

$$\partial_{(N_L, N_R)} Z(\beta_L, \beta_R)\big|_{\beta_L \beta_R = 4\pi^2} = 0 \quad \text{for any } N_L, N_R. \tag{4.2}$$

Interestingly the presence of the fixed locus gives us a new set of functional relations which contain derivatives with $N_L + N_R$ even. At $\beta_L = \beta_R = 2\pi$, the two terms in (4.2) are separately zero for $N_L + N_R$ odd, in agreement with (4.1).

Let us examine the implications of (4.1) and (4.2) at low order. For $(N_L, N_R) = (1, 0)$, we find

$$\beta_L \langle E_L \rangle + \beta_R \langle E_R \rangle\big|_{\beta_L \beta_R = 4\pi^2} = 0 \,. \tag{4.3}$$

One should remember that $\langle E_L \rangle$ and $\langle E_R \rangle$ are functions of $\beta_L$ and $\beta_R$. Evaluating at $(\beta_L, \beta_R) = (2\pi, 2\pi)$ gives us $\langle E \rangle_{\beta_L = \beta_R = 2\pi} = 0$. In fact, $\langle E_L \rangle$ and $\langle E_R \rangle$ are separately zero at $(\beta_L, \beta_R) = (2\pi, 2\pi)$. This follows from (4.1) at $(N_L, N_R) = (1, 0)$ and $(N_L, N_R) = (0, 1)$. Let us check (4.3) against AdS$_3$ gravity. The left- and right-moving energies at the phase transition line $\beta_L \beta_R = 4\pi^2$ are $\langle E_L \rangle = \frac{c(\beta_R - \beta_L)}{48\beta_L}, \langle E_R \rangle = \frac{c(\beta_L - \beta_R)}{48\beta_R}$, which together satisfy (4.3).

We can also get the behavior of $\langle E_L \rangle$ along the fixed line as $\beta_L \to 0$. For $\beta_L < \beta_R$, it is easy to check that $e^{-\beta_L E_L - \beta_R E_R}(E_L - E_R) \geq e^{-\beta_L E_R - \beta_R E_L}(E_L - E_R)$. The inequality is strict for states with $E_L \neq E_R$, which exist in any CFT spectrum. Let us now sum this inequality over the spectrum. The left-hand-side is $\langle E_L - E_R \rangle$. Using parity symmetry, the right-hand-side is $\langle E_R - E_L \rangle$. Thus,

$$\langle E_L \rangle > \langle E_R \rangle \qquad \text{if } \beta_L < \beta_R. \tag{4.4}$$

Now (4.3) says that on the fixed line, $\langle E_L \rangle = -4\pi^2 \langle E_R \rangle / \beta_L^2$. So for (4.4) to be true, $\langle E_R \rangle$ must be strictly negative for temperatures on the fixed line with $\beta_L < \beta_R$. In particular, $\langle E_R \rangle \in [-c/24, 0)$ after imposing unitarity. It is important that $\langle E_R \rangle$ cannot be zero, which follows from the strictness of the inequality in (4.4). Taking the limit $\beta_L \to 0$ on the fixed line, we see that

$$\lim_{\substack{\beta_L \to 0 \\ \beta_L \beta_R = 4\pi^2}} \langle E_L \rangle = \frac{e_L}{\beta_L^2} \qquad \text{for some } e_L \in \mathbb{R}^+. \tag{4.5}$$

Interestingly, this is the high-temperature scaling of energy with temperature in CFT, which we have shown holds in the purely left-moving sector, even though $\beta = (\beta_L + \beta_R)/2 \to \infty$.

One can derive further results about expectation values at higher orders, but these apply only to parity-invariant theories. We now turn to constraints on the spectrum of operators which will also apply to parity non-invariant theories.

## 4.2 Bounds on the spectrum of operators

Assuming $c > 1$, the partition function can be expressed as a sum over Virasoro characters, which can be written down explicitly in terms of the Dedekind eta function. The vacuum Virasoro character is

$$\chi_{\text{vac}} = \frac{\exp[\beta_L(c-1)/24]\left(1 - e^{-\beta_L}\right)}{\eta(i\beta_L/2\pi)} \frac{\exp[\beta_R(c-1)/24]\left(1 - e^{-\beta_R}\right)}{\eta(i\beta_R/2\pi)}, \tag{4.6}$$

and the character for a non-conserved Virasoro primary with weights $(h, \bar{h})$ is

$$\chi(h, \bar{h}) = \frac{\exp\{-\beta_L[h - (c-1)/24]\}}{\eta(i\beta_L/2\pi)} \frac{\exp\{-\beta_R[\bar{h} - (c-1)/24]\}}{\eta(i\beta_R/2\pi)}. \tag{4.7}$$

Let us consider the following specific linear combination of derivatives from (4.2):

$$\mathcal{A} := [\partial_{(3,0)} \chi_{\text{vac}}] \partial_{(1,0)} - [\partial_{(1,0)} \chi_{\text{vac}}] \partial_{(3,0)}, \tag{4.8}$$

where all derivatives are understood to be taken at the same point along the fixed locus $\beta_L \beta_R = 4\pi^2$. We know that $\mathcal{A}[Z] = 0$ and also, by construction, $\mathcal{A}[\chi_{\text{vac}}] = 0$. For large values of $(h, \bar{h})$, we have that $\mathcal{A}[\chi(h, \bar{h})]$ is positive. Thus, in order to have $\mathcal{A}[Z] = 0$, at least one state must exist in the spectrum for which $\mathcal{A}[\chi(h, \bar{h})]$ is negative. Let us see what we get from this particular constraint when $\beta_L = \beta_R = 2\pi$. The outcome is that there must exist a primary state within the blue shaded region of figure 2. Reference [51] established the existence of a primary between the red dashed lines in figure 2. While the upper boundary of the

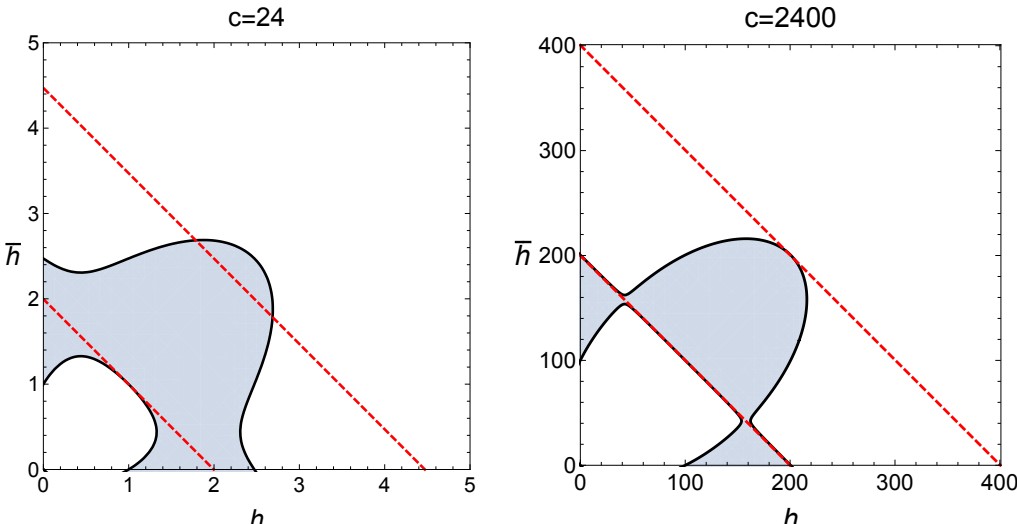

Figure 2: In any CFT, a state should exist within the blue shaded region. The dashed red lines delimit the existence region derived in [51] whose upper bound asymptotes to $h + \bar{h} = \frac{c}{6} + 0.473695$ as $c \to \infty$. Along the line $h = \bar{h} = \Delta/2$, the tip of the shaded region asymptotes to $\Delta = \frac{c}{6} + 0.951160$ at large-$c$. Interestingly, the $h$ and $\bar{h}$ intercepts of the upper boundary of the shaded region are at $c/12$ at leading order in $c$.

shaded region juts out of the upper red line at finite $c$, it is contained strictly within the red line at infinite $c$, as we can deduce from the exact expression for the boundary of the shaded region, which, in the infinite-$c$ limit, is given by:

$$\left(h + \bar{h} - \frac{c}{12}\right)\left[\frac{c}{24}\left(h + \bar{h}\right) - \left(h^2 - h\bar{h} + \bar{h}^2\right)\right] = 0 . \tag{4.9}$$

(The boundary curves develop kinks in the infinite-$c$ limit.) Thus we improve on the bound on the first excited primary in large-$c$ theories. Interestingly, the $h$ and $\bar{h}$ intercepts of the upper boundary are at $c/12$ at leading order in $c$.

Via numerical work, reference [53] improved the bound on $\Delta^{(1)}$ for $c$ of order 100 to $\Delta^{(1)} \lesssim c/9$. However, at asymptotically large $c$, the best result to compare to is still $\Delta^{(1)} < c/6$ obtained in [51]. Keeping in mind our qualitative goals, we have not tried to exhaustively apply the machinery of linear programming. We leave it for future work to see if one can do better by exhaustively searching for an optimal functional about points other than $\beta_L = \beta_R = 2\pi$.

### 4.3 Bounds on the twist gap

It is known that all unitary, modular-invariant $CFT_2$'s with $c > 1$ and no conserved currents have a sequence of Virasoro primaries with twist accumulating to $(c-1)/12$ [53, 55]. In this section we will derive a simpler statement: there must be a Virasoro primary with twist smaller than $(c-1)/12 + \epsilon$ for every $\epsilon > 0$. Recall that twist equals $2\min(h, \bar{h})$.

Let us consider the following linear combination of derivatives from (4.2):

$$\mathcal{B} := \alpha_1 \partial_{(1,0)} + \alpha_2 \partial_{(2,0)} - \partial_{(2,1)} , \tag{4.10}$$

$$\alpha_1 = \left(\frac{\beta_L}{24}\right)^2 - q\left(\frac{\beta_L}{24}\right), \qquad q \geq 2 , \tag{4.11}$$

$$\mathcal{B}[\chi_{\text{vac}}] = 0 . \tag{4.12}$$

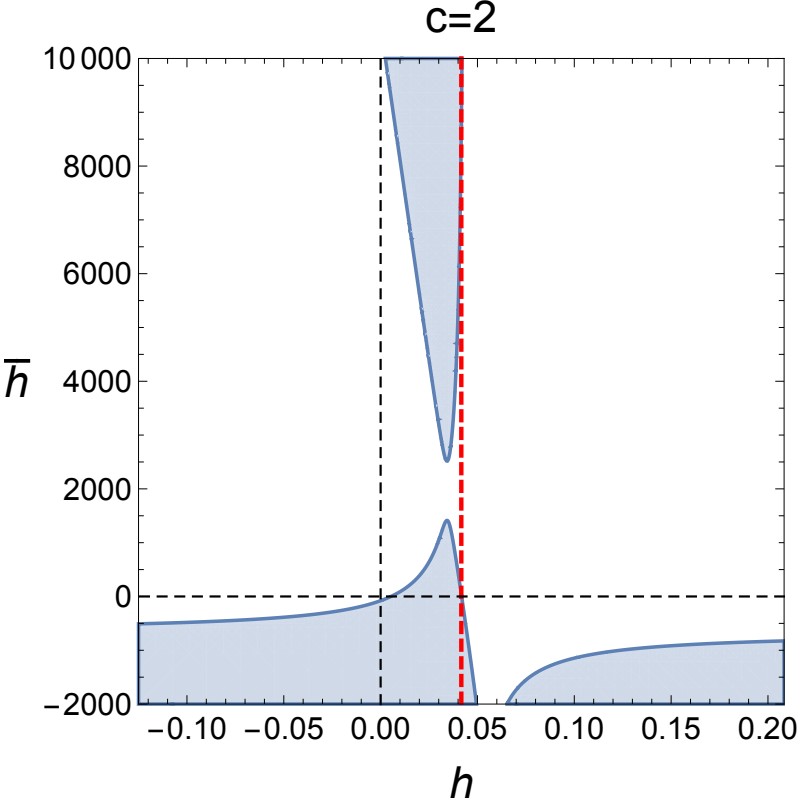

Figure 3: Deriving the twist gap. The shaded regions are where the functional (4.10) is negative, for $c = 2$. The expected upper bound $(c-1)/24$ is denoted by the dashed red line. We have taken $q = 24$ and $\beta_L = 100\pi$. There must be a state in the shaded regions with $h$ and $\bar{h}$ positive. We have shown the unphysical quadrants with negative $h$ and $\bar{h}$ to exhibit a more complete picture of the functional.

Here we pick the constant $\alpha_1$ to be as in (4.11) and we pick $\alpha_2$ to be such that the functional vanishes on the vacuum (4.12). The constant $q$ can be any real number greater than or equal to two. A sample plot of the regions in the $(h, \bar{h})$ plane where the functional is negative is given in in Figure 3. The blue shaded regions are where the functional is negative, and thus we must have a state that is in the blue region and has $h$ and $\bar{h}$ positive.

Let us summarize the important features. The right edge of the top shaded region asymptotes to a vertical line for any value of $\beta_L$. Taking larger and larger values of $\beta_L$, the vertical asymptote approaches the line $h = (c-1)/24$ from the right. The region on the bottom right is always in the unphysical region. The bottom left region intersects the horizontal axis at a value of $h$ which also approaches $(c-1)/24$ from the right. All these features are straightforward to check analytically.

Taking functionals with larger and larger values of $\beta_L$, we establish the existence of a state with $\min(h, \bar{h}) < (c-1)/24 + \epsilon$ for every $\epsilon > 0$.

Note that the existence of a continuous fixed locus and the even order derivative constraints were both crucial here. We needed to go to the limit $\beta_L \to \infty$ along the fixed locus and include the $(2, 0)$ derivative to derive this result.

# 5 Comments and extensions

## 5.1 Time reversal

One might wonder about additional discrete symmetries other than parity, and if they provide additional constraints. One natural candidate is time reversal symmetry. However time reversal does not provide new constraints on the partition function, which we now explain.

The parity transformation is usually defined as flipping the signs of *all* the spatial coordinates. When the number of spatial dimensions is even, this transformation has determinant one, and thus belongs to the part of the orthogonal group that is connected to the identity. It was emphasized in [56] that a more uniform treatment of even and odd dimensions is possible if we instead work with the *reflection* transformation $R$, which is defined to flip only one of the spatial coordinates. The determinant of this transformation is always $-1$. Importantly, in Euclidean signature, the 'time' coordinate is not special, so 'time reversal' is also simply a reflection, which is why time reversal symmetry does not provide us with new constraints.

## 5.2 Fixed locus in four-point function crossing equation

Let us consider the four point correlation function of identical scalar operators. Usually crossing is stated as the mapping $(z, \bar{z}) \to (1-z, 1-\bar{z})$ of the cross ratios, which has the unique fixed point $(z, \bar{z}) = (1/2, 1/2)$. Parity acts as $(z, \bar{z}) \to (\bar{z}, z)$. The combination of crossing and parity thus acts as $(z, \bar{z}) \to (1-\bar{z}, 1-z)$, so we see that parity again extends the fixed point to a fixed locus $z + \bar{z} = 1$.

In dimensions bigger than two, the conformal block is automatically symmetric under $z \leftrightarrow \bar{z}$. For example, one can see this explicitly in $D = 4$, where the conformal block contribution of a primary with spin $\ell$ and dimension $\Delta$ is [57,58]

$$G_{\ell,\Delta}^{(D=4)}(z, \bar{z}) = \frac{z\bar{z}}{\bar{z} - z} \left[ k_{2h-2}(z) k_{2\bar{h}}(\bar{z}) - (z \leftrightarrow \bar{z}) \right], \tag{5.1}$$

where $k_{2h}(z) := z^h \, {}_2F_1(h, h, 2h; z)$. More abstractly, we use conformal invariance to place three points at $0$, $(1, 0, \ldots, 0)$ and $\infty$, and the fourth point in the $1-2$ plane. In dimensions bigger than two, the symmetry under exchanging $z$ and $\bar{z}$ is automatic because of the existence of a rotation in the $2-3$ plane that maps $z$ to $\bar{z}$.[2] Thus, the fixed point $(z, \bar{z}) = (1/2, 1/2)$ is always enhanced to the fixed line $z + \bar{z} = 1$ in dimensions bigger than two.

In two dimensions, there is no third dimension that we can use to rotate $z$ to $\bar{z}$, and therefore the $z \leftrightarrow \bar{z}$ symmetry is not automatic. This is expected given the many examples of perfectly good CFT$_2$'s which have left-right asymmetry. In more detail, the 2D global conformal block of a primary with quantum numbers $(h, \bar{h})$ is (again considering identical external scalars)

$$G_{h,\bar{h}}^{(D=2)}(z, \bar{z}) = k_{2h}(z) k_{2\bar{h}}(\bar{z}). \tag{5.2}$$

Note the manifest asymmetry with respect to interchange of $z$ and $\bar{z}$. To get something parity symmetric, we have to add the contribution of the parity-transformed state with the same OPE coefficient. The four point function in a parity invariant CFT$_2$, therefore has the following conformal block decomposition:

$$\langle \phi(x_1)\phi(x_3)\phi(x_3)\phi(x_4) \rangle = \frac{1}{x_{12}^{2\Delta_\phi}} \frac{1}{x_{34}^{2\Delta_\phi}} \sum_{\mathcal{O}} f_{\phi\phi\mathcal{O}}^2 \left[ k_{2h}(z) k_{2\bar{h}}(\bar{z}) + k_{2h}(\bar{z}) k_{2\bar{h}}(z) \right]. \tag{5.3}$$

Now one can see explicitly that $G(z, \bar{z}) = G(\bar{z}, z)$. Thus, combining crossing and parity, we conclude that there is a fixed line $z + \bar{z} = 1$.

---

[2] We thank Douglas Stanford for discussions on this point.

### 5.3 Theories with an internal $U(1)$ symmetry

Consider a relativistic $CFT_2$ with an internal $U(1)$ symmetry. Let us note immediately that not every theory with a $U(1)$ symmetry admits an extra 'charge conjugation' symmetry. However, what is true by the existence of antiparticles is that in any relativistic $CFT_2$, for every state with quantum numbers $\{h, \overline{h}, q, \overline{q}\}$ there is a state with quantum numbers $\{h, \overline{h}, -q, -\overline{q}\}$.[3]

Introducing the left- and right-moving chemical potentials $z$ and $\overline{z}$, the flavored partition function is defined as

$$Z(\tau, \overline{\tau}, z, \overline{z}) := \text{Tr}\left[ e^{2\pi i \tau(L_0 - c/24)} e^{-2\pi i \overline{\tau}(\overline{L}_0 - c/24)} e^{2\pi i z Q} e^{-2\pi i \overline{z} \overline{Q}} \right]. \tag{5.4}$$

Here $Q$ and $\overline{Q}$ are the left- and right-moving $U(1)$ charge operators with eigenvalues $q$ and $\overline{q}$, respectively. The existence of antiparticles implies $Z(\tau, \overline{\tau}, z, \overline{z}) = Z(\tau, \overline{\tau}, -z, -\overline{z})$. Modular transformations act as [59–63]

$$Z(\tau, \overline{\tau}, z, \overline{z}) \mapsto e^{-\pi i k(z^2/(c\tau+d)+\overline{z}^2/(c\overline{\tau}+d))} Z\left( \frac{a\tau+b}{c\tau+d}, \frac{a\overline{\tau}+b}{c\overline{\tau}+d}, \frac{z}{c\tau+d}, \frac{\overline{z}}{c\overline{\tau}+d} \right). \tag{5.5}$$

Note that this transformation automatically builds in the relation $Z(\tau, \overline{\tau}, z, \overline{z}) = Z(\tau, \overline{\tau}, -z, -\overline{z})$: choosing $a = d = -1$ and $b = c = 0$ simply flips the signs of $z$ and $\overline{z}$. The action of modular $S$-inversion is

$$S : Z(\tau, \overline{\tau}, z, \overline{z}) \mapsto e^{-\pi i k(z^2/\tau + \overline{z}^2/\overline{\tau})} Z\left( -\frac{1}{\tau}, -\frac{1}{\overline{\tau}}, \frac{z}{\tau}, \frac{\overline{z}}{\overline{\tau}} \right). \tag{5.6}$$

As usual this has fixed point $\tau = -\overline{\tau} = i$, $z = \overline{z} = 0$ on the real section $\overline{\tau} = \tau^*$. This was used by [59, 60, 64] in the modular bootstrap.

Now we consider an $SP$ transformation to see if we have a fixed line in the presence of these chemical potentials:

$$SP : Z(\tau, \overline{\tau}, z, \overline{z}) \mapsto e^{-\pi i k(-\overline{z}^2/\overline{\tau} + z^2/\tau)} Z\left( \frac{1}{\overline{\tau}}, \frac{1}{\tau}, \frac{\overline{z}}{\overline{\tau}}, \frac{z}{\tau} \right). \tag{5.7}$$

The fixed locus is given by $\tau\overline{\tau} = 1$ and $\overline{z}\tau = z$. Notice that the latter constraint also makes the anomalous prefactor vanish. In terms of real parameters $\tau = \tau_1 + i\tau_2$ and $z = z_1 + iz_2$, we have a two-dimensional fixed plane in $\mathbb{R}^4$ given by

$$\tau_1 = \frac{z_1^2 - z_2^2}{z_1^2 + z_2^2}, \quad \tau_2 = \frac{2z_1 z_2}{z_1^2 + z_2^2}. \tag{5.8}$$

In analogy with the unflavored case considered in this paper, we expect that this fixed locus can be used to improve the results of [59, 60, 64] for $CFT_2$'s with a $U(1)$ symmetry.

## Acknowledgments

We would like to thank Nathan Benjamin, Scott Collier, Thomas Hartman, Simeon Hellerman, Ying-Hsuan Lin, Shu Heng Shao, Xi Yin and Sasha Zhiboedov for useful conversations. TA is supported by the Natural Sciences and Engineering Research Council of Canada, and by grant 376206 from the Simons Foundation. RM is supported by US Department of Energy grant No. DE-SC0016244. ES supported in part by NSF grant no. PHY-1316748 and Simons Foundation grant 488643.

---

[3]We emphasize that this is not in any way a 'charge conjugation' symmetry. The existence of antiparticles follows from Lorentz invariance in two and higher dimensions, via the CPT theorem. Interestingly, there is no CPT theorem in one dimension, and consequently, this statement is not true in one dimension. We thank A. Liam Fitzpatrick, Guy Gur-Ari, Daniel Harlow, Nathan Seiberg and Edward Witten for discussions on this point.

# A  Definitions

The grand canonical ensemble is parameterized by the inverse-temperature $\beta$ and angular potential $\theta$. In two-dimensional CFTs they are often packaged into

$$2\pi\tau := i\beta_L := \theta + i\beta \,, \qquad 2\pi\overline{\tau} := -i\beta_R := \theta - i\beta \,. \tag{A.1}$$

As is customary, we will treat $\tau$ and $\overline{\tau}$ as a priori independent complex variables. When trading variables, this then translates into treating $(\beta_L, \beta_R)$ or $(\beta, \theta)$ as independent sets of complex variables. Given these definitions it is easy to see that

$$\beta = \frac{\beta_L + \beta_R}{2} \,. \tag{A.2}$$

Introduce the variables $q$ and $\overline{q}$ via

$$q := \exp(2\pi i\tau), \qquad \overline{q} := \exp(-2\pi i\overline{\tau}) \,. \tag{A.3}$$

The partition function is defined as a function from $\mathbb{C}^2$ to $\mathbb{C}$:

$$Z(\tau, \overline{\tau}) := \mathrm{Tr}\left[ q^{L_0 - c/24} \; \overline{q}^{\overline{L}_0 - \overline{c}/24} \right], \tag{A.4}$$

where $c$ and $\overline{c}$ are the central charges for the holomorphic and the anti-holomorphic Virasoro algebras. One can show the partition function is modular invariant with respect to $SL(2, \mathbb{Z})$ transformations on $\tau$ and $\overline{\tau}$, without restricting to the section $\overline{\tau} = \tau^*$: see [32] for an argument using the Weierstrass preparation theorem.

States are labeled by a pair of non-negative real numbers $(h, \overline{h})$, which are the eigenvalues of the $L_0$ and $\overline{L}_0$. The conformal dimension $\Delta$ and spin $s$ of a state are then defined as

$$\Delta := h + \overline{h} \,, \qquad s := h - \overline{h} \,. \tag{A.5}$$

The twist of a state is defined as

$$t = 2\min(h, \overline{h}). \tag{A.6}$$

In the literature, states are also frequently parametrized either by their left- and right-moving energies, defined as:

$$E_L := h - \frac{c}{24} \,, \qquad E_R := \overline{h} - \frac{\overline{c}}{24} \,, \tag{A.7}$$

or by their total energy $E$ and angular momentum $J$, defined as:

$$E := \Delta - \frac{c + \overline{c}}{24} \,, \qquad J := s - \frac{c - \overline{c}}{24} \,. \tag{A.8}$$

In the main text, we will only consider theories with $c = \overline{c}$.

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
