# Peer review of "Parity and the modular bootstrap"

_SciPost Physics, doi:SciPost Phys. 5, 022 (2018)_

## Round 2 · Referee Report · Anonymous · 2018-6-25

Strengths

1-Elegant argument that allows to extend previous results.
2-Results fill in gaps in previous papers.
3-Short and clearly written.

Weaknesses

1-The methods used are not genuinely new, but rather an application of known methods to a combined new symmetry.

Report

This paper considers a combined parity and modular transformation of 2d CFTs. This symmetry allows to apply bootstrap techniques to obtain new results. Namely, the universality of the phase diagram of 2d CFTs with sparse spectrum is established for general spin potentials. It also improves the bound on the lightest operator in a large $c$ theory. These are both nice additions to previous results in the literature.

Requested changes

none

---

## Round 2 · Referee Report · Anonymous · 2018-7-10

Strengths

1. The authors used clear and concise arguments to argue qualitative properties of 2d CFT. $\\$

2. It is a nice idea to utilize a continuous line of fixed points to bound theories. $\\$

3. The paper is well-written and clarifies several potential confusions along the way.

Weaknesses

1. Parts of the paper do not feel complete, even for the purpose of demonstrating qualitative features. For example, the right side of Figure 2 shows two sharp kinks at 3 derivative order. One may thus wonder about the shape of the region at higher orders, and about whether such kinks persist. Such a qualitative question can be answered without going to too high an order. $\\$

2. Almost all argued properties were already known in the literature. The reduction of the allowed region in Figure 2 was phrased as a "novel constraint" in the abstract; however, the reduction had not been explicitly mapped out in the past mainly due to the lack of a clear target statement worth pursuing. A new revealing statement is also lacking in the present paper.

Report

This paper explores how a continuous line of fixed points in parity-invariant CFT$_2$ may be used to generate bounds and demonstrate universal features. Their major achievement is to prove a previous conjecture about the universality of the spectrum in large c theories. Other utilizations of the continuous line of fixed points are also discussed, and some are further generalized to theories without parity symmetry.

Requested changes

1. Above (3.8), I suppose the authors want to keep $\beta_L$ and $\beta_R$ BOTH fixed, with $\beta_L \beta_R \neq 4\pi^2$. $\\$

2. I think the statement "This shows..." below (3.8) follows more from (3.7) than (3.8). Suppose $Z_H = 100 Z_{H^c}$, one would not say that $Z_{H^c}$ is dominating, but (3.8) would remain true. $\\$

3. Below (3.12), it is commented that "If $\sqrt{E_L E_R} > c/24$, we can do the integral by saddle point." Isn't one still able to perform a saddle point approximation even when $\sqrt{E_L E_R} > c/24$, except that now the saddle value depends on the details of $Z$? $\\$

4. I think it should be made clearer that Section 3.2 is just a review. $\\$

5. Does the proof in Section 4.3 only apply to parity-invariant theories? Relatedly, does the last sentence in the abstract need certain qualifiers? $\\$

6. I personally find Figure 3 not very illuminating. I would rather they write up their analytic argument following (4.12) in more explicit detail.

  • validity: top
  • significance: good
  • originality: good
  • clarity: high
  • formatting: perfect
  • grammar: perfect

Author:  Tarek Anous  on 2018-08-02  [id 302]

(in reply to Report 2 on 2018-07-10)

Dear Referee, thank you for taking the time to assess our paper. We will now try to address your comments.

Weaknesses:

In reference to weakness 1) we plan to clarify figure 2. To do so, we will modify it from simply showing the upper bound on the operator dimension to showing the entire region (including now a lower bound), where an operator must exist as dictated by modular invariance. We have changed the caption accordingly and have added a parenthetical in the ensuing description to indicate that the sharp features are only a result of the large-c limit and are not related to the low derivative order.

It is incumbent upon us to address weakness 2) as it would seem that the referee is suggesting that most of our results are not novel. However, the only previously known result from our paper is the twist gap, for which we provide an alternate derivation. The continuous fixed line as a practical tool for the numerical/analytic modular bootstrap is a new idea. The proof of the HKS conjecture and the reduction of the allowed region in Figure 2, both of which form the primary results of the paper, are also new. In summary, we respectfully disagree with the referee's commentary and contend that the majority of our results are indeed novel.

Requested changes:

1) We will include the word ``both'' in the line above equation (3.8) in order to clarify that we must keep both temperatures fixed.

2) If $Z_H = 100 Z_{H^c}$, we would indeed still say that the non-heavy states dominate the free energy, because considering their contribution alone gives the correct large-$c$ answer. In other words, $\log(100) = O(1)$ in large-$c$ counting, while $\log Z_{H^c} = O(c)$. We therefore intend to leave this part unchanged.

3) In section 3.2 we will reword the discussion to indicate that we can always do the integral by saddle point.

4) It is our opinion that it would be a mischaracterization to label section 3.2 as review. The precise reason one might be interested in proving that CFTs have a universal free energy at large $c$ is that this property explains why the Cardy formula extends to states beyond its naive regime of validity. Since our paper manages to finally show that CFTs have a universal free energy at large $c$, for arbitrary angular potentials, section 3.2 stands as a direct consequence of this result.

5) The proof in section 4.3 extends to parity non-invariant theories. This is because it is a statement about the twist, defined as the $2\text{min}(h,\overline{h})$ and is not a bound on the individual $(h,\overline{h})$. Phrasing it as a bound on the twist makes the result hold for theories without parity symmetry, despite the fact that figure 3 would seem to suggest that we are bounding $h$.

We will change the last line of the abstract to emphasize this point.

6) While we agree with the referee that the description after (4.12) is not as explicit as it could be, this was done in order to preserve clarity rather than the opposite. The reason is that applying $\mathcal{B}$ to a generic partition function results in a nasty and cumbersome set of equations involving Dedekind eta functions and derivatives thereof. We quote the limits of these functions, which the reader is free to check, but we feel that writing down explicit equations would more likely bog the reader down given their sheer length.

We would again like to thank you for taking the time to assess and improve our paper.

Tarek Anous, Raghu Mahajan, and Edgar Shaghoulian

---

## Round 3 · Referee Report · Anonymous (Referee 2) · 2018-8-16

Strengths

1-The authors used clear and concise arguments to argue qualitative properties of 2d CFT.$\\$

2-It is a nice idea to utilize a continuous line of fixed points to bound theories.$\\$

3-The paper is well-written and clarifies several potential confusions along the way.

Weaknesses

1-Most techniques are existing ones, and some results are re-derivations of old results.

Report

This paper explores how a continuous line of fixed points in parity-invariant CFT2 may be used to generate bounds and demonstrate universal features. Their major achievement is to prove a previous conjecture about the universality of the spectrum in large c theories. Other utilizations of the continuous line of fixed points are also discussed, and some are further generalized to theories without parity symmetry.

Requested changes

1-Following up on Weakness 1 in my earlier report, I find the comment below (4.9) about the kink in the infinite c limit slightly misleading to the reader who is less familiar with bootstrap. Here there are two limits involved, infinite derivative order and infinite c. The physical order of limit is first infinite derivative order, and then infinite c. So the physical meaning of a kink that develops in the infinite c limit at 3 derivative order is unclear. I suggest that the authors add a footnote explaining this point, saying that the kink may be an artifact of finite derivative order. Better yet, if they wish, the authors may go to 5 or 7 derivatives to check the fate of the kink, and make a more definitive statement.$\\$

Note: There has also been no change or response regarding Changes 2, 4 and 6 in my earlier report, but since they were subjective comments, I leave those points to the authors' liberty.

  • validity: top
  • significance: good
  • originality: good
  • clarity: high
  • formatting: perfect
  • grammar: perfect

Author:  Tarek Anous  on 2018-08-30  [id 311]

(in reply to Report 1 on 2018-08-16)
Category:
remark
answer to question

Dear Referee,

Thank you again for your report. We would like to stress that the plots we have shown do not come from an exhaustive optimization up to a certain order of derivatives, as is usually the case in the bootstrap. They simply present the constraint that arises from a single combination of derivatives, with a judiciously chosen set of coefficients. In particular, the meaning of "the shape at higher orders" is, in not a well-defined notion in our presentation.

If we had chosen to perform a systematic optimization, one should expect a shape similar to the one that appears in our paper, but then the question of how the shape changes as we increase derivative order can be addressed, in that context. We instead judged that such a systematic optimization was outside the scope of our paper, opting instead for a qualitative illustration.

Without undertaking a systematic optimization, the best one can do to address your comment is to keep the combination of derivatives fixed and vary the central charge. As seen by the plots, the kinks only develop at large central charge.

As a result we believe it is best to leave the section unchanged.

Thank you for taking the time to assess our paper.

Tarek Anous, Raghu Mahajan, Edgar Shaghoulian

---

## Round 3 · Author Response

We would like to thank the referees for their kind assessment of our paper. We hope that we have addressed their concerns in our replies and in the list of changes describing this resubmission.

---

## Round 3 · List of Changes

1) In order to clarify which results are independent of the parity symmetry, we have changed the last line of the abstract to: ``Our results for the free energy and constraints on the operator spectrum extend to theories without parity symmetry through the construction of an auxiliary parity-invariant partition function.''

2) We have included the word ``both'' in the line above equation (3.8) in order to clarify that we must keep both temperatures fixed.

3) In section 3.2 we have reworded the discussion to indicate that we can always do the integral by saddle point. We have added the sentence ``If $\sqrt{E_{L}\,E_R}>c/24$, this saddle point is at a value of $\beta_L$ and $\beta_R$ such that $\beta_L \beta_R < 4\pi^2$.'' to explain the significance of that condition in the final equation for the Cardy formula.

4) We have modified figure 2 from simply showing the upper bound on the operator dimension to showing the entire region (including now a lower bound), where an operator must exist as dictated by modular invariance. We have modified the relevant section and caption accordingly and have added a parenthetical in the ensuing description to indicate that the sharp features are only a result of the large-c limit and are not related to the low derivative order.

5) We have fixed a typo in the original submitted version of our paper. Previously it was written that $\tilde{\rho}\leq 2\rho(h,\overline{h})$. We have replaced this with the correct expression (3.12):$\tilde{rho} \leq 2\, \text{max}\left[\rho(h, \overline{h}),\rho(\overline{h},h)\right]$.

---

## Editorial Decision

published